# Unicompartmental Knee Replacement in Obese Patients: A Systematic Review and Meta-Analysis

**DOI:** 10.3390/jcm10163594

**Published:** 2021-08-15

**Authors:** Stefano Campi, Giuseppe Francesco Papalia, Carlo Esposito, Erika Albo, Francesca Cannata, Biagio Zampogna, Rocco Papalia, Vincenzo Denaro

**Affiliations:** 1Department of Orthopaedic and Trauma Surgery, Campus Bio-Medico University of Rome, 00128 Rome, Italy; s.campi@unicampus.it (S.C.); carlo.esposito@unicampus.it (C.E.); e.albo@unicampus.it (E.A.); b.zampogna@unicampus.it (B.Z.); r.papalia@unicampus.it (R.P.); denaro@unicampus.it (V.D.); 2Unit of Endocrinology and Diabetes, Department of Medicine, Campus Bio-Medico University of Rome, 00128 Rome, Italy; f.cannata@unicampus.it; 3Multi-Specialist Clinical Institute for Orthopaedic Trauma Care (COT), 98124 Messina, Italy

**Keywords:** unicompartmental knee replacement, obesity, body mass index, revisions, infections

## Abstract

Thanks to modern surgical techniques and implants, traditional exclusion criteria for unicompartmental knee arthroplasty (UKA) are no longer considered contraindications. The aim of this study is to clarify the impact of obesity on functional outcomes and revision rates of UKA. We performed a comprehensive systematic review using PubMed–Medline, Google Scholar and Cochrane Central. Then, we extracted data related to body mass index (BMI), age and follow-up, functional outcome scores and rate of revisions (all-cause, aseptic and septic). Patients were stratified according to BMI into two groups: non-obese (BMI < 30) and obese (BMI ≥ 30). We identified 22 eligible studies, of which 13 were included in the meta-analysis. Patients with a BMI > 30 had a significantly higher likelihood for revision (*p* = 0.02), while the risk of septic revision was similar (*p* = 0.79). The clinical outcome measures showed a significant difference in favor of patients with a BMI < 30 (*p* < 0.0001). The improvements in Oxford Knee Score and Knee Society Score were significant in both obese and non-obese patients, although the latter showed inferior results. The results of this systematic review and meta-analysis show that BMI is not a contraindication to UKA. However, obese patients have a higher risk for aseptic failure and lower improvement in clinical scores compared to non-obese patients.

## 1. Introduction

Unicompartmental knee arthroplasty (UKA) is a valid treatment for end-stage knee osteoarthritis (OA), affecting a single femoro-tibial compartment [1]. The popularity of unicompartmental knee replacement (UKR) has increased as excellent functional outcomes and survival have been reported in long-term follow-up studies. UKA has important advantages compared to total knee arthroplasty (TKA), including lower intraoperative blood loss and risk of transfusion [2,3] as well as accelerated recovery [4]. In addition, UKA is associated with a decreased length of stay in hospital, lower readmission rates [5], lower infection rates and fewer major medical complications, such as thromboembolism, stroke and myocardial infarction, compared to patients undergoing TKA [6]. Some authors have reported that UKA produces more natural knee biomechanics and healing of physiological gait pattern [7], with superior patient-reported clinical and functional outcomes [8,9,10,11]. Despite these advantages, data from national joint registries showed a higher risk or revision in patients undergoing UKA [12]. Correct patient selection is paramount to achieve good outcomes for UKA, reducing the risk of UKA failure and revision surgery. According to the Kozinn and Scott criteria proposed in 1989, body weight over 82 kg is a contraindication to UKA [13]. More recently, it has been demonstrated that many of the “traditional” contraindications to UKA are not necessary, including a high BMI [14,15,16]. However, the role of BMI and its influence on the results of UKA and TKA is still being debated. Over the last few decades, the number of obese patients needing treatment for end-stage knee arthritis has significantly increased. Body weight has been shown as a modifiable risk factor for knee osteoarthritis and disease progression [17,18,19]. In addition, adverse events such as dislocation, aseptic loosening, superficial and deep infection and revision surgery are more common in obese patients undergoing TKA [20,21,22]. In contrast, the impact of obesity on the results of UKA is still unclear, with some surgeons offering UKA to both obese and non-obese patients, while others consider a high BMI as a contraindication and a reason of concern for potential early failure. The aim of this systematic review and meta-analysis is to compare the results of obese and non-obese patients in terms of clinical and functional scores and risk of revision. Our hypothesis is that a higher BMI would be associated with lower functional outcome scores and higher risk of septic and aseptic failures.

## 2. Materials and Methods

Two independent reviewers (S.C. and C.E.) performed a systematic review of the literature according to the Appendix A guidelines (preferred reporting items of systematic reviews and meta-analysis).

### 2.1. Information Sources and Search

An electronic search was performed through PubMed–Medline database, Google Scholar and Cochrane Central using different combinations of keywords; “unicompartmental”, “unicondylar”, “partial knee arthroplasty”, and “UKA” were combined with each of the keywords “BMI”, “Body Mass Index”, “obesity”, “weight” and “survival”, “complications”, “outcome”. Other sources were searched by reviewing reference lists and citing manuscripts to identify relevant studies missed during the electronic search. We limited our search to the English language literature.

### 2.2. Eligibility Criteria and Study Selection

The following study types were considered for inclusion: controlled randomized trials, case control studies, case series, retrospective case series and prospective case series. We followed the PICO strategy (Appendix B), and we included studies that evaluated the clinical and functional outcomes and the rate of aseptic and septic revisions in obese and non-obese patients who underwent UKA. Only articles published after 2000 were considered for inclusion. Studies before 2000 or reporting on old implant designs were excluded from the review. Only articles with BMI were included in the study, as weight alone does not necessarily indicate obesity. The last search was performed on 20 April 2021. A total of 916 studies were reviewed by title and/or abstract to determine study eligibility based on inclusion criteria. Duplicate references were discarded. The title and abstract of the articles identified through the search were read. Then, the full text of the remaining articles was read by two reviewers (S.C. and C.E.). We included studies that meet at least one of the following inclusion criteria: (1) BMI index used to stratify levels of obesity, (2) outcome reported with a validated scoring system, (3) failures and revisions reported. Studies with a mean follow-up shorter than 2 years were excluded. When more studies presented the same group of patients, we selected only the most recent one. All the abstracts of the studies that met the inclusion criteria were independently evaluated, and full texts were retrieved and assessed to prove eligibility.

### 2.3. Data Collection Process

Screening was performed in two phases to identify relevant titles, abstracts and full texts. Two reviewers (S.C. and C.E.) collected and summarized the data in tables using Microsoft Office Excel (2013 version, Microsoft Corporation, Redmond, WA, USA). Data extraction was performed independently to reduce the risk of bias. In the case of discrepancy, the data extraction was repeated and discussed. The following data were extracted: number of patients, number of knees (UKA procedures), mean and range for BMI, age and follow-up, rate of revision surgical procedures (all-cause, aseptic, and septic), and functional outcome scores. As clinical outcomes, we selected the Knee Society Score (KSS) and its subscores (knee, function and objective) to evaluate both the knee prosthesis function and patients’ functional abilities after knee arthroplasty. Oxford Knee Score (OKS) was used as a patient-reported outcome to assess function and pain in activities of daily living after knee replacement. Range of motion (ROM) was assessed to establish the knee flexion and extension before and after UKA.

### 2.4. Methodological Quality Assessment

After the level of evidence (LOE) of the studies had been assessed according to the Oxford criteria, the risk of bias of each study was assessed with the Methodological index for non-randomized studies (MINORS) score [23], which includes 8 items to assess the risk of bias in non-comparative studies and a further 4 items for comparative studies.

### 2.5. Statistical Analysis

Data analysis was performed using Review Manager software 5.3 (RevMan 2014). Continuous outcomes were used to compare function scores in patients with a BMI above or below 30. Dichotomous outcomes were used to compare the rate of revisions between the two groups. Continuous data were shown as mean difference (MD), with 95% confidence intervals. Dichotomous outcomes were shown as odds ratio (OR), with 95% confidence intervals. For the evaluation of the weight of the samples of the included studies, the number of revisions per year of follow-up were used instead of the total number of revisions. Our meta-analysis was stratified by BMI category (non-obese BMI < 30 and obese BMI ≥ 30). We quantitatively pooled the risk ratios for all-cause revision and for septic revision using a random effects model with 95% confidence intervals (CIs). Heterogeneity was assessed using the I^2^ test. A fixed-effect model was used for data with low heterogeneity (I^2^ < 55%); otherwise, a random-effect model was performed. Results were significant at *p* < 0.05.

## 3. Results

### 3.1. Literature Search

The PRISMA flowchart for study selection is shown in Figure 1. The literature search resulted in a total of 916 references. After abstract evaluation, 875 papers were excluded due to duplication (26) or being off-topic (849). After full-text evaluation, 16 further papers were excluded because they did not meet the inclusion criteria or reported incomplete data. Three studies [15,24,25] reported the results of the same cohort (or similar cohorts) of patients. When present in the same analysis, only the study with the longest follow-up was considered. Therefore, 22 papers were included in the final systematic review: 12 of these studies were retrospective studies [26,27,28,29,30,31,32,33,34,35,36,37], 5 were prospective studies [38,39,40,41,42] and 5 were case series [25,43,44,45,46]. Thirteen studies showed adequate information on revisions and functional outcomes to be included in the meta-analysis.

### 3.2. Patient Demographics

Patient demographics for each study are summarized in Table 1. Eleven studies [25,26,27,28,29,30,33,35,38,39,42] provided mean age, mean BMI and mean follow-up time for all BMI subgroups. Other studies reported mean age, BMI or follow-up time for the whole study population and not for each BMI subgroup. Two studies [43,45] only considered patients with BMI > 40 and BMI > 30, respectively. One study [28] divided patients according BMI but did not report the mean BMI of each subgroup. One study [31] did not report the number of procedures, mean age or mean BMI but only the division of patients according to BMI and rate of revision. The reported follow-up periods ranged from a minimum of 2 years to a maximum of 12 years.

### 3.3. Clinical Outcome

The clinical outcome collected for each study can be found in Table 2. Five studies [27,29,33,37,39,40] reported clinical outcome using the Knee Society Score (KSS) knee and function. These authors reported in their studies the average pre- and post-operative score. Two studies [35,46] reported clinical outcome using the KSS function and KSS pain. Two studies [41,42] used KSS function and KSS objective. In these last studies, not all reported average pre- and post-operative score, as detailed in Table 2. The Oxford Knee Score (OKS) was used to evaluate clinical outcome in six studies [25,27,29,38,39,42]. One study [29] described for all patient groups an improvement in the post-operative OKS, but in the reported table the post-operative value was lower than the pre-operative one. For this reason, we excluded the OKS of this study from the meta-analysis. One study [38] reported data from two institutes: OKS at center 1 and Objective and Functional KSS score at centers 1 and 2. Data from center 1 was already used in a study of UKR on patients with weight above or below 82 kg, and data from center 2 was used in a similar study with BMI above or below 32 [15,16]. Seven studies [27,32,37,39,40,42,45] reported the clinical outcome using the ROM. All of these reported a good or excellent mean post-operative ROM for all BMI subgroups.

### 3.4. Failures and Revisions

Survival rate, revision rate and cause of revision are described in Table 3. Not all studies reported survival rate or distinguished revision causes by patient BMI subgroup.

### 3.5. Methodological Evaluation

The MINORS score for the included studies ranged from 7 to 14, with a mean value of 11. Therefore, the methodological quality was heterogeneous between the different studies. More precisely, the worst item regarded the unbiased assessment of the study endpoint, while stated aims, inclusion of consecutive patients and appropriate endpoints were at low risk of bias in almost all studies (Table 4).

### 3.6. Effect of Intervention

The meta-analysis performed a comparison between patients with a BMI < 30 and with a BMI > 30 for functional outcomes and revision rates. Eight studies [25,27,29,37,38,39,40,42] analyzed the clinical outcomes after UKR between obese and non-obese patients (Figure 2). OKS was significantly higher in patients with a BMI < 30 compared to those with a BMI > 30 (MD 3.81, 95% CI, 2.06 to 5.56, *p* < 0.0001). The KSS knee showed better improvements in non-obese patients, but no significant differences (MD 2.15, 95% CI, −0.60 to 4.89, *p* = 0.13). KKS function increased significantly after UKA in non-obese group (MD 6.61, 95% CI, 1.50 to 11.72, *p* = 0.01). Finally, evaluating all the reported clinical outcomes, a significant difference was shown in favor of patients with a BMI < 30 compared to patients with BMI > 30 (MD 4.38, 95% CI, 2.28 to 6.48, *p* < 0.0001). Moreover, 11 studies [27,28,29,31,33,38,39,40,41,42,44] analyzed the revisions after UKA and showed a significantly increased likelihood for revision in patients with a BMI > 30 (OR 1.42, 95% CI, 1.05 to 1.92, *p* = 0.02) (Figure 3). Instead, the rate of septic revisions did not show significant differences between the two groups (OR 0.90, 95% CI, 0.41 to 1.97, *p* = 0.79) (Figure 4).

## 4. Discussion

The present study demonstrated an increased risk of revision for all causes in obese patients (BMI ≥ 30) undergoing UKA compared with non-obese patients (BMI < 30). There was no significant difference in the incidence of revision for infection between the two groups (*p* = 0.79). We found significant differences in post-operative clinical outcomes in non-obese patients compared with obese patients. However, obese and non-obese patients experienced similar improvements in OKS and KSS knee and function, suggesting that all patients undergoing UKA benefit from the procedure, regardless of BMI. Our results are comparable with those of previous meta-analyses on the effect of BMI on the results of UKA. Van der List et al. [47] studied the influence of different patients characteristics on the outcome of UKA, including age (young vs. old), gender (male vs. female), BMI (obese vs. non-obese), presence of patellofemoral osteoarthritis and status of the anterior cruciate ligament. The author found no significant differences in the outcomes of obese versus non-obese patients (OR 2.06; *p* = 0.11). Moreover, the analysis of six cohort studies and two registries comparing revision rates in 21,204 patients showed a slightly higher likelihood for revision in obese patients, without a statistically significant difference (OR, 0.71; *p* = 0.09). A further study conducted by Agarwal et al. [48] demonstrated no statistically significant difference following UKA between obese and non-obese patients in overall complication rates (*p* = 0.52), infection rates (*p* = 0.81), and revision surgeries (*p* = 0.06). Moreover, the authors did not find differences for revisions specifically for infection (*p* = 0.71) or aseptic loosening (*p* = 0.75). Therefore, they proved that obesity did not lead to poorer post-operative outcomes following UKA and should not be considered a contraindication for UKA. In addition, Musbahi et al. [49] in their meta-analysis showed that the mean revision rate of obese patients (BMI > 30) was 0.33% per annum higher than that of non-obese patients; however, this difference was not statistically significant (*p* = 0.82). In a meta-analysis by Chaudhry et al. [50] on TKA, the risk ratios for all-cause revision surgical procedures were 1.19 (*p* = 0.02) in severely obese (BMI > 35 kg/m^2^), 1.93 (*p* < 0.001) in morbidly obese (BMI > 40 kg/m^2^), and 4.75 (*p* < 0.001) in super-obese (BMI > 50 kg/m^2^) patients compared to patients with a normal BMI. They also demonstrated an increased risk of septic revision surgical procedures in severely obese (risk ratio 1.49; *p* < 0.001), morbidly obese (risk ratio 3.69; *p* < 0.001) and super-obese (risk ratio 4.58; *p* = 0.04) patients. Moreover, they proved that there was no higher risk of other causes of revision (i.e., aseptic revisions) among patients with a BMI of >35 kg/m^2^, regardless of BMI. Furthermore, they showed no significant difference in the improvement in functional outcomes in patients with severe or morbid obesity compared with non-obese patients; functional outcome change scores were 0.06 lower (*p* = 0.44) in severely obese, 0.06 lower (*p* = 0.45) in morbidly obese, and 0.52 lower (*p* < 0.001) in super-obese patients. Comparing these results to those of our study, compared to TKA, UKA showed similar effects on the clinical outcome but a lower increase in the risk of failure in obese patients. Accordingly, BMI should be considered as a risk factor for revision after knee replacement surgery; however, this risk is significantly higher in patients undergoing TKA compared to those receiving UKA. There are several hypotheses to justify these results. First, obese patients are likely to perform less physical activity than non-obese patients, therefore leading to minor use of their implant; reduced physical activity compensates for the increased load of the obese patients in terms of prosthesis survival. Some authors have suggested that the follow-up of most studies is not long enough to observe an increased revision rate in obese patients. Finally, some suggest that the use of a mobile bearing implant design reduces the risk of revision in obese patients by facilitating better load distribution. The goal of this study is to focus and synthesize existing evidence related to the outcomes of UKA in patients with obesity. We aimed to develop evidence-based decision making for clinicians and surgeons in order to better quantify specific risks and benefits for patients. Based on these results, further studies are needed to deepen the current conclusions, analyze the correlation with patient comorbidities and evaluate surgical interventions that can improve outcomes in obese patients. Furthermore, our results demonstrate that reducing access to UKR for patients with a high BMI needs to be critically re-evaluated. There were several limitations to our study. First, our meta-analysis was based on the quality of the included studies, which obtained an average value of 11 out of 16 according to MINORS criteria. The reason for the low quality can be attributed to the small cohort and retrospective design of most studies, which made this study subject to accuracy of record and biases inherent to this study type. Second, in the meta-analysis, studies were selected on the basis of a uniform cutoff value (BMI >/< 30 kg/m^2^), and only studies reporting both groups were included. However, only including comparative studies reduced the risk for bias. Third, during our literature research, we found relatively few studies that analyzed the super-obese group, resulting in greater imprecision in the reported point estimate. Therefore, further prospective studies are needed to evaluate outcomes in super-obese and morbidly obese patients. Fourth, the mean times to the revisions were rarely described, making it impossible to determine whether revisions were required in the short, intermediate, or long term. Fifth, another limitation was the selection of outcome measure. We selected those that were more relevant to the decision making, i.e., revision surgical rate and clinical scores. However, obesity and morbid obesity are often associated with medical comorbidities that may independently affect outcomes. We didn’t consider important confounding variables that could influence risk of infection (e.g., patient comorbidities such as hypertension, hyperlipidemia, and diabetes; operative time; wound-healing complications; use dosage and timing of perioperative antibiotics) because they were infrequent across the studies. Moreover, we did not adjust the metanalysis by age of the patients, comorbidities or severity of knee OA. For this reason, our results must be confirmed with analyses adjusted for relevant confounding factors. Moreover, the surgical procedures were carried out by different surgeons, often in the same study, who could have diverse indications for surgery in patients with unicompartmental knee OA. This could have introduced operator-dependent variability. Recent studies show that hospitals and surgeons with low surgical volumes had higher failure rates compared to hospitals and surgeons that performed UKA more regularly. Therefore, the overall revision rate might also be influenced by this phenomenon. Finally, in our study we left out possible differences in terms of outcome score and rate of revision between medial and lateral replacements, and fixed and mobile-bearing UKA designs.

## 5. Conclusions

Our systematic review and meta-analysis demonstrated that the risk of revision was greater in obese patients (BMI > 30). However, the difference was lower than reported by similar studies on TKA. The risk of revision for infection in patients with a BMI > 30 was not significantly higher than that of non-obese patients. Although the improvements in OKS and KSS function were statistically significant for patients with a BMI < 30, obese and non-obese patients experienced similar improvements after UKA. Therefore, this meta-analysis suggests that all patients undergoing UKA benefit from the intervention, regardless of BMI. Accordingly, BMI should not be considered as a contraindication for UKA. However, obese patients should be informed about the increased risk of failure and inferior functional outcome of joint replacement surgery and should lose weight prior to undergoing surgery.

## Figures and Tables

**Figure 1 jcm-10-03594-f001:**
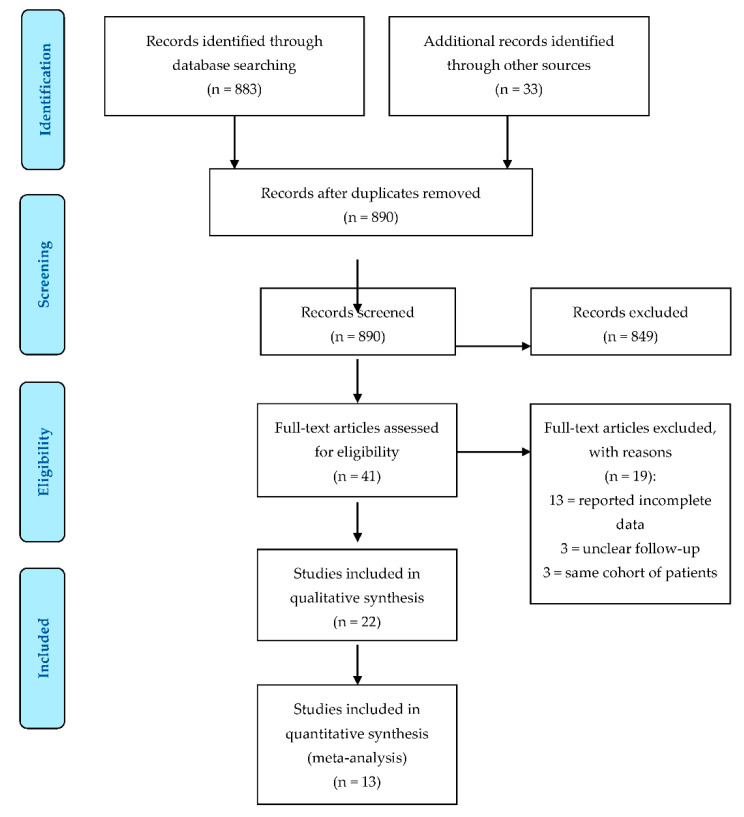
Preferred Reporting Items for Systematic Review and Meta-Analysis (PRISMA) flow diagram.

**Figure 2 jcm-10-03594-f002:**
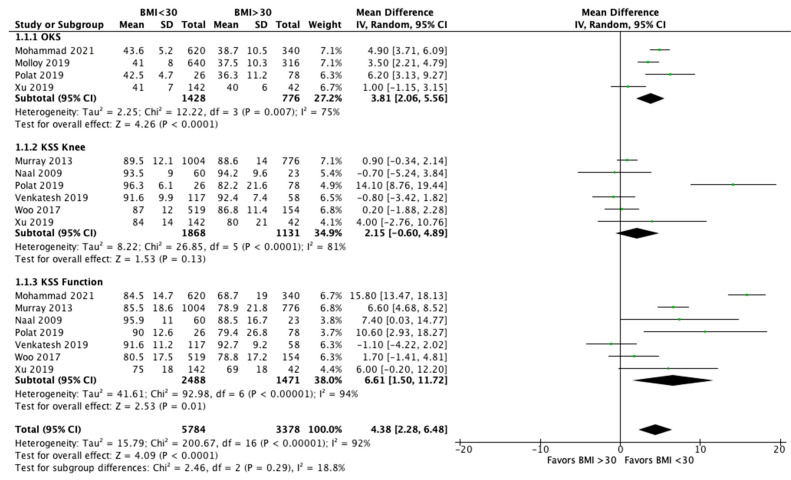
Clinical outcomes.

**Figure 3 jcm-10-03594-f003:**
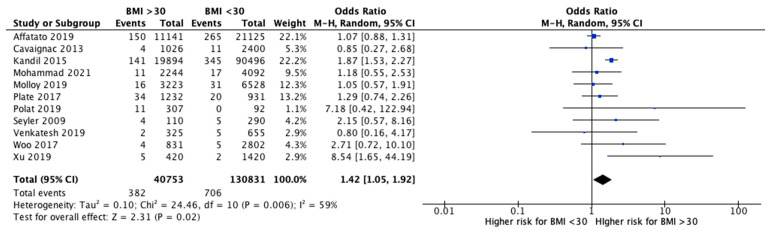
Revisions.

**Figure 4 jcm-10-03594-f004:**
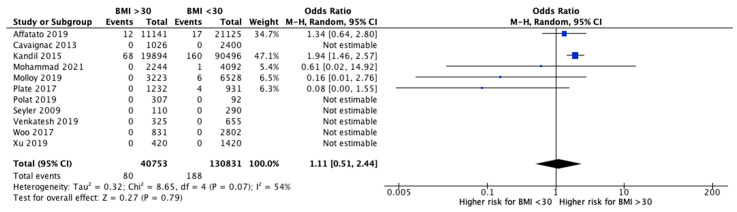
Septic revisions.

**Table 1 jcm-10-03594-t001:** Patient demographics.

Study	Year	Study Design	LOE	Cohort	Patients	Number of UKA	Mean Age, Years (Range)	Mean BMI, kg/m^2^ (Range)	Mean Follow-Up (Range)
Nettrour et al.	2019	RS	II	Not Morbidly Obese (BMI < 40)	81	101	57.6 ± 8.3 (40–83)	33.1 ± 5 (20–39)	3.5 ± 1.3 years (2–6.8)
				Morbidly Obese (BMI ≥ 40)	71	89	55.3 ± 9.1 (40–79)	45.8 ± 5.6 (>40)	3.2 ± 1.1 years (2–6.8)
Polat et al.	2019	RS	II	Normal and Overweight (BMI < 30)	86	26	61.5 ± 7.3	27.3 ± 2.3	42.7 ± 14.1 months
				Obese (BMI = 30–34.9)		40	60.5 ± 7.7	32.7 ± 1.5	40.6 ± 13.5 months
				Morbidly Obese (BMI ≥ 35)		38	59.0 ± 7.1	40.9 ± 5.6	53.9 ± 12.7 months
Seth et al.	2019	CS	IV	Morbidly Obese (BMI ≥ 40)	103	121	58 (43–75)	43 (40–51)	7 years (2 months–15 y)
Molloy et al.	2019	PS	III	Normal (BMI < 25)	202	207	70.3 ± 10	22.6 ± 3	10.2 years (5–16)
				Overweight (BMI = 25–29.9)	427	433	66.4 ± 10	27.3 ± 1	
				Obese (BMI = 30–34.9)	218	220	64.9 ± 9	32.1 ± 1	
				Morbidly Obese (BMI ≥ 35)	94	96	61.7 ± 8	39.0 ± 4	
Affatato et al.	2019	RS	III	Normal (BMI < 30)	3976	3250	67.8 (24–90)	NR	6.5 years (0–16.3)
				Obese (BMI = 30–39.9)		1636	65.7 (28–89)	NR	
				Morbidly Obese (BMI ≥ 40)		78	61.2 (47–79)	NR	
Xu et al.	2019	PS	I	Control (BMI < 30)	142	142	62.4 ± 7.8	25.6 ± 2.9	minimum 10 years
				Obese (BMI≥ 30)	42	42	56.5 ± 6.4	33.4 ± 3	
Venkatesh et al.	2019	PS	I	BMI < 30	148	117	61.7 (44–80)	29.2 kg/m^2^ (21–38)	5.6 years (2–10)
				BMI ≥ 30		58			
Plate et al.	2017	CS	IV	Underweight (BMI < 18.5)	672	1	64 ± 11	32.1 ± 6.5	34.6 ± 7.8 months
				Normal (BMI = 18.5–24.9)		91			
				Overweight (BMI = 25–29.9)		229			
				Obese (BMI = 30–34.9)		227			
				Severely Obese (BMI = 35–39.9)		115			
				Morbidly Obese (BMI = 40–44.9)		42			
				Super Obese (BMI ≥ 45)		41			
Woo et al.	2017	RS	II	Normal (BMI <25)	230	230	65 ± 8	22.6 ± 1.8	5.4 years (2.5–8.5)
				Overweight (BMI = 25–29.9)	289	289	62 ± 8	27.4 ± 1.3	
				Obese (BMI = 30–34.9)	124	124	61 ± 8	31.9 ± 1.4	
				Severely Obese (BMI = 35–39.9)	30	30	58 ± 9	38.5 ± 3.6	
Zengerink et al.	2015	RS	II	Not Obese (BMI < 30)	122	63	60.0 (± 8.1)	26.9 (± 2.3)	3.9 years (2.0–12.2)
				Obese (BMI ≥ 30)		64	60.9 (± 6.6)	33.6 (± 3.2)	5.1 years (2.0–10.8)
Kandil et al.	2015	RS	II	Non-Obese (BMI < 30)	12,928	NR	NR	NR	7 years
				Obese (BMI = 30–39.9)	1823	NR	NR	NR	
				Morbidly Obese (BMI ≥ 40)	1019	NR	NR	NR	
Cepni et al.	2014	CS	IV	BMI > 30	67	67	61 ± 7.3	35.7 ± 2.6	67.5 months ± 15.4
Murray et al.	2013	CS	IV	Normal (BMI < 25)	2438	378	69 (38–91)	23 (15–24.9)	4.6 years (1–12)
				Overweight (BMI = 25–29.9)		856	65 (33–89)	27	
				Obese (BMI = 30–34.9)		712	61 (34–88)	32	
				Severely Obese (BMI = 35–39.9)		286	61 (34–87)	37	
				Morbidly Obese (BMI = 40–44.9)		126	58 (41–87)	42	
				Super Obese (BMI ≥ 45)		80	59 (41–78)	50 (45–69)	
Thompson et al.	2013	RS	II	BMI < 35	173	229	66 (33–89)	29.3 (18.4–48.7)	2 years
				BMI ≥ 35	32				
Cavaignac et al.	2013	RS	II	Not Obese (BMI < 30)	254	200	66.5 (39–92)	27 (19–29)	12 years (7–22)
				Obese (BMI ≥ 30)		90	65.8 (55–84)	34 (30–43.2)	11.4 years (7–17)
Xing et al.	2012	RS	II	BMI < 30	140	178	67 (36–90)	28.8 (19.7–48.5)	54 months (24–77)
				BMI = 30–34.9					
				BMI = 35–39.9					
				BMI ≥ 40					
Bonutti et al.	2011	RS	II	Not Obese (BMI < 35)	33	40	68 (48–79)	28 (23–34)	3 years (2–7)
				Obese (BMI ≥ 35)	34	40	65 (45–81)	38 (35–47)	3 years (2–6)
Kuipers et al.	2010	RS	II	BMI < 30	437	437	62.8 (39.3–84.6)	30.1 (17.7–47.3)	2.6 years (0.1–7.9)
				BMI ≥ 30					
Seyler et al.	2009	PS	IV	Not Obese (BMI < 30)	68	58	72 (44–91)	27 (17–39)	60 months (24–68)
				Obese (BMI ≥ 30)		22			
Naal et al.	2009	RS	II	Normal (BMI = 18.5–24.9)	77	13	66 (46–84)	27.8 (20.2–39.2)	2 years
				Overweight (BMI = 25–29.9)		47			
				Obese (BMI = 30–34.9)		23			
Berend et al.	2005	CS	IV	Not Obese (BMI < 32)	61	73	66.3 (43–83)	31.65 (19–50)	40 months (24–69)
				Obese (BMI ≥ 32)					
Mohammad et al.	2021	PS	I	Normal (BMI = 18.5–24.9)	756	186	69.1 ± 10.4	23.2 ± 1.4	6.6 years (5–10) ± 2.7
				Overweight (BMI = 25–29.9)		434	66.5 ± 10.1	27.5 ± 1.4	
				Obese Class 1 (BMI = 30–34.9)		213	64.6 ± 9.4	32.2 ± 1.4	
				Obese Class 2 (BMI ≥ 35)		127	63.6 ± 8.6	38.3 ± 3.5	

RS: retrospective study; CS: case series; PS: prospective studies.

**Table 2 jcm-10-03594-t002:** Clinical outcome.

Study	Cohort	KSS		KSS Knee		KSS Function		KSS Objective		OKS		ROM (°)	
		Pre	Post	Pre	Post	Pre	Post	Pre	Post	Pre	Post	Pre	Post
Polat	Normal and Overweight (BMI < 30)			44.0 ± 4.3	96.3 ± 6.1	35.8± 22.2	90 ± 12.6			11.4 ± 7.8	42.5 ± 4.7	116.3 ± 12.0	128.3 ± 19.3
	Obese (BMI = 30–34.9)			42.1 ± 11.8	88.8 ± 10.8	32.3 ± 21.2	87.8± 12.4			11.4 ± 8.0	39.3 ± 7.2	106.9 ± 11.2	124.5 ± 11.8
	Morbidly Obese (BMI ≥ 35)			43.9 ± 9.8	75.2 ± 27.2	36.6 ± 13.5	70.5 ± 34			15.1 ± 7.0	33.1 ± 13.6	113.2 ± 12.5	117.4 ± 18.3
Molloy	Normal (BMI < 25)									26.1 ± 10	40.6 ± 8		
	Overweight (BMI = 25–29.9)									25.5 ± 9	41.2 ± 8		
	Obese (BMI = 30–34.9)									23.3 ± 8	36.6 ± 11		
	Morbidly Obese (BMI ≥ 35)									22.2 ± 9	39.5 ± 8		
Xu	Control (BMI < 30)			47 ± 18	84 ± 14	60 ± 17	75 ± 18			28 ± 7	41 ± 7	123 ± 17	127 ± 13
	Obese (BMI ≥ 30)			44 ± 20	80 ± 21	60 ± 14	69 ± 18			27 ± 8	40 ± 6	116 ± 15	116 ± 13
Woo	Normal (BMI < 25)			44 ± 18	87 ± 12	62 ± 17	81 ± 18			32 ± 8	17 ± 5		
	Overweight (BMI = 25–29.9)			43 ± 17	87 ± 12	61 ± 16	80 ± 17			33 ± 8	18 ± 5		
	Obese (BMI = 30–34.9)			46 ± 18	88 ± 10	60 ± 16	80 ± 17			33 ± 7	18 ± 5		
	Severely Obese (BMI = 35–39.9)			33 ± 17	82 ± 15	54 ± 16	74 ± 17			37 ± 6	20 ± 6		
Zengerink	Not Obese (BMI < 30)		29.2 ± 11.4										
	Obese (BMI ≥ 30)		27.9 ± 10.8										
Cepni	BMI > 30									18.5 ± 4.7	40 ± 5	117.6 ± 5	127 ± 5.2
Murray	Normal (BMI < 25)						84 ± 18.5 */85± 17.8 **		86 ± 9.9 */94 ± 8.8 **	27 ± 9.2	42 ± 6.8		
	Overweight (BMI = 25–29.9)						86 ± 19.1 */87 ± 18.8 **		85 ± 10 */92 ± 13.8 **	25 ± 8.5	41 ± 7.5		
	Obese (BMI = 30–34.9)						80 ± 21.2 */81 ± 20.9 **		84 ± 14.9 */91 ± 13.3 **	23 ± 7.9	39 ± 8.9		
	Severely Obese (BMI = 35–39.9)						69 ± 27.4 */79 ± 23.1 **		82 ± 13.6 */91 ± 12.3 **	19 ± 5.9	39 ± 9.3		
	Morbidly Obese (BMI = 40–44.9)						79 ± 21.1 */76 ± 20.8 **		93 ± 6.1 */91 ± 14.7 **	19 ± 8.4	39 ± 7.7		
	Super Obese (BMI ≥ 45)						76 ± 13.6 */73 ± 24 **		84 ± 19.8 */89 ± 13.7 **	23 ± 6.2	41 ± 3.7		
Scott	BMI < 35	53 ± 20	81 ± 22									117	124
	BMI ≥ 35												
Cavaignac	Not Obese (BMI < 30)				80 (70–96)		85 (50 -100)						
	Obese (BMI ≥ 30)				78 (64–90)		77 (50–100)						
Bonutti	Not Obese (BMI < 35)					49 (20–70)	97 (80–100)						
	Obese (BMI ≥ 35)					50 (30–70)	95 (70–100)						
Seyler	Not Obese (BMI < 30)					48 ± 9	92 ± 7	49 ± 9	95 ± 4				
	Obese (BMI ≥ 30)												
Naal	Normal (BMI = 18.5–24.9)	132 ± 24.5	190.5 ± 13.7	57.5 ± 14.8	95.1 ± 4.7	74.5 ± 16.8	95.4 ± 9.7					123.7 ± 12.2	128.1 ± 5.2
	Overweight (BMI = 25–29.9)		189.1 ± 14.8		93.1 ± 9.8		96 ± 11.4						129.9 ± 7.1
	Obese (BMI = 30–34.9)		182.7 ± 23.4		94.2 ± 9.6		88.5 ± 16.7						125.7 ± 6.6
Berend	Not Obese (BMI < 32)	54 (34–92)	87			48 (20–90)	63						
	Obese (BMI ≥ 32)												
Venkatesh	BMI < 30			47.4 ± 5.5	91.6 ± 9.9	55.3 ± 4.6	91.6 ± 11.2					111.3 ± 11.7	118.4 ± 11.8
	BMI ≥ 30			46.2 ± 5.6	92.4 ± 7.43	54.9 ± 4.6	92.7 ± 9.2					108.7 ± 10.1	118.3 ± 12.1
Mohammad	Normal (BMI < 25)					71.9 ± 14.8	80.9 ± 16	61.9 ± 16.8	90.5 ± 12.1	26.9 ± 8.4	42 ± 5.6		133.9 ± 10.8
	Overweight (BMI = 25–29.9)					73.8 ± 16.8	86.0 ± 13.9	62.3 ± 14.5	95.6 ± 4.8	26.7 ± 7.5	44.3 ± 4.9		128.5 ± 9.7
	Obese (BMI = 30–34.9)					68.9 ± 16.9	67.9 ± 11.8	59.2 ± 15.8	87.5 ± 17.5	24.2 ± 8.5	40.1 ± 9.7		126.4 ± 5
	Severely Obese (BMI = 35–39.9)					63.7 ± 17.8	70 ± 27.1	53.9 ± 13.8	83.1 ± 16.4	20.8 ± 8.8	36.4 ± 11.4		125.6 ± 9.1

* Center 1, ** Center 2.

**Table 3 jcm-10-03594-t003:** Failures and revisions.

Study	Cohort	Survival Rate	Number of Revision (%)	Causes of Failure, Reoperation
Nettrour	Not Morbidly Obese (BMI < 40)	NR	6 (6%)	Minor procedures-aseptic: 2 (2%)
				Lateral/anterior compartment progression: 1 (1%)
				Loose tibial component: 2 (2%)
				Infection: 1 (1%)
	Morbidly Obese (BMI ≥ 40)	NR	19 (21.3%)	Minor procedures-aseptic: 3 (3.4%)
				Lateral/anterior compartment progression: 7 (7.8%)
				Bearing instability: 5 (5.6%)
				Loose tibial component: 2 (2.2%)
				Infection: 2 (2.2%)
Polat	Normal and Overweight (BMI < 30)	NR	0	-
	Obese (BMI = 30–34.9)	NR	3 (27%)	Tibial + femoral loosening: 3
	Morbidly Obese (BMI ≥ 35)	NR	8 (72.7%)	Tibial loosening: 3
				Tibial + femoral loosening: 3
				Tibial component collapse: 2
Seth	Morbidly Obese (BMI ≥ 40)	91.7% at 2 years, 86.3% at 5 years	19	Improper patient selection: 1
				OA progression: 4
				Issue in technique: 9
				Unexplained pain: 2
				Aseptic loosening of tibial component: 2
				Traumatic liner dislocation: 1
Molloy	Normal (BMI < 25)	92% at 10 years	13 (6.3%)	OA progression: 26
				Unexplained pain: 7
	Overweight (BMI = 25–29.9)	95% at 10 years	18 (4.2%)	Bearing dislocation: 7
				Infection: 6
	Obese (BMI = 30–34.9)	94% at 10 years	10 (4.5%)	Aseptic loosening: 2
				Instability: 1
	Morbidly Obese (BMI ≥ 35)	93% at 10 years	6 (6.3%)	Malposition: 1
				ACL injury: 1
				Unknown: 1
Affatato	Normal (BMI < 30)	92.6% at 5 years, 87.4% at 10 years	265 (8.1%)	Total aseptic loosening: 121
				Pain without loosening: 53
				Tibial aseptic loosening: 35
				Septic loosening: 17
				Femoral aseptic loosening: 16
				Insert wear: 12
				Breakage of prosthesis: 7
				Dislocation: 4
	Obese (BMI = 30–39.9)	91.4% at 5 years, 86.7% at 10 years	145 (8.8%)	Total aseptic loosening: 55
				Pain without loosening: 41
				Tibial aseptic loosening: 27
				Septic loosening: 12
				Femoral aseptic loosening: 1
				Insert wear: 1
				Breakage of prosthesis: 3
				Dislocation: 5
	Morbidly Obese (BMI ≥ 40)	95.5% at 5 years, 87.5% at 10 years	5 (6.4%)	Total aseptic loosening:2
				Pain without loosening:1
				Tibial aseptic loosening:1
				Dislocation:1
Xu	Control (BMI < 30)	98.6% at 10 years	2	OA progression: 2
	Obese (BMI ≥ 30)	88.1% at 10 years	5	OA progression: 2
				Subsidence of tibial component: 2
				Polyetilene wear:1
Plate	Underweight (BMI < 18.5)	NR	0–0	Revision to TKA: Persistent knee pain (46%), Unknown (21%), Tibial component loosening (12%), Progression of DJD to adjacent compartment (9%), Tibial component subsidence (7%), Infection (5%)
	Normal (BMI = 18.5–24.9)		2 (2.2%)–1 (1.1%)	
	Overweight (BMI = 25–29.9)		14 (6.1%)–3 (1.3%)	
	Obese (BMI = 30–34.9)		13 (5.7%)–4 (1.8%)	Conversion from InLay to OnLay: Tibial component subsidence (46%), Tibial component loosening (27%), Persistent knee pain (9%), Undersized tibial component (9%), Infection (9%)
	Severely Obese (BMI = 35–39.9)		10 (8.7%)–2 (1.7%)	
	Morbidly Obese (BMI = 40–44.9)		4 (9.5%)–0	
	Super Obese (BMI ≥ 45)		0–1 (2.4%)	
Woo	Normal (BMI < 25)	NR	1	Subsidence: 1
	Overweight (BMI = 25–29.9)		4	OA progression: 3
				Persisiting pain: 1
	Obese (BMI = 30–34.9)		2	OA progression: 2
	Severely Obese (BMI = 35–39.9)		2	OA progression: 1
				Fracture: 1
Zengerink	Not Obese (BMI < 30)	87%	18	Unexplained pain: 8
				OA progression: 2
				Instability: 3
				Aseptic loosening: 2
	Obese (BMI ≥ 30)			Traumatic loosening of tibial component: 1
				Atraumatic migration of tibial component: 1
				Unknown reason: 1
Kandil	Non-Obese (BMI < 30)	NR	345 (2.7%)	Major complications: 303 (2.3%)
				Minor complications: 532 (4.1%)
				Local complications: 439 (3.4%)
				Medical complications: 256 (2.0%)
	Obese (BMI = 30–39.9)		84 (4.6%)	Major complications: 97 (5.3%)
				Minor complications: 179 (9.8%)
				Local complications: 68 (3.7%)
				Medical complications: 142 (7.8%)
	Morbidly Obese (BMI ≥ 40)		57 (5.6%)	Major complications: 73 (7.2%)
				Minor complications: 132 (13%)
				Local complications: 68 (6.7%)
				Medical complications: 106 (10.4%)
Cepni	BMI > 30	95.6% at 5 years	3	Insert dislocation: 3
Murray	Normal (BMI < 25)	97.6% at 5 years, 94.9% at 10 years	9	Unexplained pain: 3
				Infection: 2
				OA progression: 2
				Aseptic loosening: 1
				Bearing dislocation: 1
	Overweight (BMI = 25–29.9)	96.8% at 5 years, 93% at 10 years	25	Unexplained pain: 7
				Aseptic loosening: 5
				Infection: 4
				OA progression: 3
				Bearing dislocation: 3
				Traumatic ACL rupture: 1
				AVN of lateral femoral condyle: 1
				Fracture: 1
	Obese (BMI = 30–34.9)	95.3% at 5 years, 95.3% at 10 years	18	Unexplained pain: 6
				Aseptic loosening: 5
				OA progression: 3
				Bearing dislocation: 3
				Periprothetic fracture: 1
	Severely Obese (BMI = 35–39.9)	93.8% at 5 years, 93.8% at 10 years	7	Aseptic loosening: 4
				Unexplained pain: 1
				Infection: 1
				Bearing dislocation: 1
	Morbidly Obese (BMI = 40–44.9)	95.2% at 5 years	4	Aseptic loosening: 2
				Unexplained pain: 1
				Infection: 1
	Super Obese (BMI ≥ 45)	100% at 5 years	0	-
Thompson	BMI < 35 BMI ≥ 35	NR	8 (3.5%)	OA progression: 2
				Tibial plateau fracture: 2
				Persistent pain: 2
				Subsidence of tibial component: 1
				Malposition of components: 1
Cavaignac	Not Obese (BMI < 30)	92% at 10 years	11	Aseptic tibial loosening: 3
				OA progression: 4
				Polyethylene wear: 1
				Unexplained pain: 1
				Impingement with LCM: 1
				Impingement with intercondylar eminence: 1
	Obese (BMI ≥ 30)	94% at 10 years	4	OA progression: 3
				Unexplained pain: 1
Xing	BMI < 30	96.2%	6 (3.8%)	Implant loosening: 3
	(BMI = 30–34.9)			Persisiting pain: 1
	BMI = 35–39.9			OA progression: 2
	BMI ≥ 40			
Bonutti	Not Obese (BMI < 35)	88%	5	Progression of OA: 2
				Tibial component loosening: 2
				Intractabile pain: 1
	Obese (BMI ≥ 35)	100%	0	
Kuipers	BMI > 30 BMI ≥ 30	84.7% at 5 years	45 (10.3%)	Persisiting pain: 13
				Aseptic loosening: 12
				OA progression: 9
				Recurrent luxation of meniscal bearing: 4
				Deep infection: 2
				Periprosthetic fracture: 3
				Traumatic instability of MCL: 1
				Malpositioning of tibial component: 1
Seyler	Not Obese (BMI < 30)	92% at 5 years, 84% at 10 years	5	Aseptic loosening: 2
				Patellofemoral/lateral pain: 3
	Obese (BMI ≥ 30)		4	Polyethylene wear: 2
				Progression of OA: 1
				Tibial plateau fracture: 1
Naal	Normal (BMI <25)	NR	3 (3.6%)	Loosening of the tibial component: 1
	Overweight (BMI = 25–29.9)			Loosening of the femoral component: 1
	Obese (BMI ≥ 30)			Intractabile pain: 1
Berend	Not Obese (BMI < 32) Obese (BMI ≥ 32)	78% at 3 years	16	Deep infection: 2 (2.7%)
				Tibial plateau fracture: 3 (4.1%)
				Intractabile pain: 4 (5.5%)
				Progression of OA: 1 (1.4%)
				Aseptic loosening: 6 (8.2%)
Venkatesh	BMI < 30	96% at 10.9 years	5 (4.27%)	Unexplained pain: 2
				Loosening of component: 2
				Polyethylene wear: 1
	BMI ≥ 30		2	Unexplained pain: 2
Mohammad	Normal (BMI < 25)	97.3% at 10 years	4	Bearing dislocation: 1
				Tibial avascular necrosis: 1
				Disease progression: 1
				Lateral meniscal tear: 1
	Overweight (BMI = 25–29.9)	96.2% at 10 years	13	Bearing dislocation: 4
				Disease progression: 3
				Suspected infection: 1
				Pain: 2
				Loose body: 1
				Sweling: 1
				Wound dehiscence: 1
	Obese (BMI = 30–34.9)	94.8% at 10 years	9	Bearing dislocation: 3
				Pain: 4
				Femoral component loosening: 1
				Disease progression: 1
	Severely Obese (BMI = 35–39.9)	98.3% at 10 years	2	Lateral tibial fracture: 1
				Disease progression: 1

**Table 4 jcm-10-03594-t004:** MINORS score.

Study	Year	A Clearly Stated Aim	Inclusion of Consecutive Patients	Prospective Collection of Data	Endpoints Appropriate to the Aim of the Study	Unbiased Assessment of the Study Endpoint	Follow-Up Period Appropriate to the Aim of the Study	Loss to Follow-Up Less Than 5%	Prospective Calculation of the Study Size	Total
Mohammad	2021	2	2	2	2	0	2	2	1	13
Nettrour	2019	2	2	1	2	0	2	2	1	12
Polat	2019	2	2	2	2	1	1	2	0	12
Seth	2019	2	2	0	2	1	1	2	0	10
Molloy	2019	2	2	1	2	0	2	2	1	12
Affatato	2019	2	2	0	2	0	2	2	0	10
Xu	2019	2	2	1	2	1	2	2	2	14
Venkatesh	2019	2	2	1	2	0	2	2	1	12
Plate	2017	2	2	1	2	0	1	2	1	11
Woo	2017	2	2	1	2	0	1	2	1	11
Zengerink	2015	2	2	0	2	0	1	1	1	9
Kandil	2015	2	2	0	2	0	2	2	2	12
Cepni	2014	2	1	0	1	0	1	2	0	7
Thompson	2013	2	2	0	2	0	1	2	0	9
Murray	2013	2	2	0	2	0	2	2	2	12
Cavaignac	2013	2	2	1	2	0	2	2	2	13
Xing	2012	2	2	1	2	0	1	1	0	9
Bonutti	2011	2	2	1	2	0	1	2	2	12
Kuipers	2010	2	2	0	2	1	1	2	2	12
Seyler	2009	2	2	2	2	0	1	2	1	12
Naal	2009	2	2	1	2	0	1	1	1	10
Berend	2005	2	2	0	2	0	1	1	1	9

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
