# Peer review of "Unicompartmental Knee Replacement in Obese Patients: A Systematic Review and Meta-Analysis"

_jcm, 2021, doi:10.3390/jcm10163594_

Round 1

Reviewer 1 Report

Dear researchers, 

I had the opportunity to revise your manuscript, aiming to clarify the impact of obesity on functional outcomes and the revision rate of UKA.

You made a great effort by identifying published communication and summarized the data, but tables are hard to read then, the analysis turns complex. 

I will suggest the following changes:

  • Table 1. List on the tables the 13 communications included in the analysis. The rest of the communication does not help with your main objective. Make sure that table headed is on all the pages. 
  • Figure 1, please remove the sentence "qualitative analysis" as you don't describe the qualitative analysis in your methods (this is a compilation of data). 
  • Table 2. Please, focus only on those variables with the potential to be analyzed in the metanalysis: OKS, KKS Knee, KSS function, Failure (revision), infection. The rest of the variables, such as VAS, KKS, KSS pain, KSS objective, and ROM, could be indexed as supplementary material or deleted. Make sure that table headed is on all the pages.
  •  Please, include a definition and interpretation of each outcome variable in the methodology section. 
  • Would you consider the option to estimate the pre-post change (delta) of each outcome variable and use this delta as another outcome variable? With this, you can compare if the change-magnitude is equal between the obese and non-obese groups.
  • Table 3: Causes of failure/reoperation could be re-categorized as: septic failure, aseptic failure, OA progression, and pain.
  • Discussion, line 270: How do you conclude this sentence? "Furthermore, the risk of revision for infection was higher in obese patient group (0,96% vs 1,12%)." A statistical test is needed. Please consider including Figure 4 for infections. 
  • Add a paragraph including limitations by not adjust the metanalysis by age, comorbidities, severity and other variables that could influence your results. 

Thank you for your effort and wish you success in this and future academic work.

Author Response

Thank you for your suggestions.

Reviewer 2 Report

This is an interesting and well written manuscript. The review is well conducted. I would simplify the tables because they are not so easy to read.

Author Response

Thank you for your suggestions.

Reviewer 3 Report

The present manuscript is a  comprehensive systematic review and meta-analysis of the orthopedic literature to determine whether the degree of obesity (overweight, obese and morbidly obese) influenced functional outcome scores and the risk of revision following UKA.

It is an interesting systematic review that has some aspects to be checked before it can be published:

Minor:

Avoid acronyms in the abstract, line 23.

Major:

Revise throughout the manuscript the obesity numbers e.g. BMI > 30 or BMI < 30, should put BMI ≥ 30... revise the symbols and categories throughout the manuscript (Line 19, 269, line 323, line 327... etc.

Could the authors please add the PICO question, just with the search terms it is more complicated to replicate it.

Figure 1

Identification: Could they indicate how many articles existed in each of the databases, i.e. how many in Pumed, how many in Google scholar and how many in Crochane?

Eligibility: The reasons for exclusion have not been indicated.

Revise the results, the authors report data on variables that are not covered in Methods, e.g. on the number of surgeons.

Have the authors  not analysed publication bias?

Discussion

I feel that the authors should revise the wording of the discussion and rewrite it completely, instead of discussing the results it seems that the authors are summarising the main previous systematic reviews and results of other studies. I think it is better to analyse well what this review adds to the previously published literature and emphasise on the results found.

All the best in your submission!

Author Response

Thank you for your suggestions.

Round 2

Reviewer 1 Report

Thank you for the changes made in the manuscript. 
I found the tables too long and difficult to understand. Please be aware that the headed line must appear on all pages.  
Success with your work!

Reviewer 3 Report

Please, could the authors  add the PICO question?

Author Response

Thank You for your suggestion. 

We inserted "Appendix A" with the PICO strategy.